# Pharmacokinetic and Pharmacodynamic Analysis of the Oxacephem Antibiotic Flomoxef against Extended-Spectrum β-Lactamase-Producing Enterobacterales from Dogs

**DOI:** 10.3390/ijms25021105

**Published:** 2024-01-16

**Authors:** Mizuki Kusumoto, Makoto Jitsuiki, Tomoki Motegi, Kazuki Harada

**Affiliations:** 1Laboratory of Veterinary Internal Medicine, Tottori University, Minami 4-101, Koyama-Cho, Tottori-shi, Tottori 680-8550, Japan; 2Joint Graduate School of Veterinary Sciences, Tottori University, Minami 4-101, Koyama-Cho, Tottori-shi, Tottori 680-8550, Japan; 3Department of Veterinary Clinical Pathobiology, Graduate School of Agricultural and Life Sciences, The University of Tokyo, Yayoi 1-1-1, Bunkyo-Ku, Tokyo 113-0032, Japan; a-t.motegi@g.ecc.u-tokyo.ac.jp

**Keywords:** pharmacokinetics–pharmacodynamics approach, flomoxef, extended-spectrum β-lactamases, Enterobacterales, dogs

## Abstract

Flomoxef (FMX) may be a potential alternative to carbapenems for dogs infected with Enterobacterales-producing extended-spectrum β-lactamase (ESBL-E). However, the appropriate dosage of FMX in dogs with ESBL-E infections has yet to be established. This study was carried out to establish appropriate treatment regimens for FMX against ESBL-E infections in dogs using a pharmacokinetics–pharmacodynamics (PK–PD) approach. Five dogs were intravenously administered at a bolus dose of FMX (40 mg/kg body weight). Serum concentrations of FMX were calculated with high-performance liquid chromatography-tandem mass spectrometry, and then applied to determine PK indices based on a non-compartmental model. The cumulative fraction of response (CFR) was estimated based on the dissemination of minimum inhibitory concentrations among wild-type ESBL-E from companion animals. From the results, the dosage regimens of 40 mg/kg every 6 and 8 h were estimated to attain a CFR of >90% for wild-type isolates of ESBL-producing *Escherichia coli*, *Klebsiella pneumoniae*, and *Proteus mirabilis* for dogs. By contrast, all regimens had a CFR of <80% for ESBL-producing *Enterobacter cloacae*. Our results indicated that dosage regimens of 40 mg/kg FMX every 6 and 8 h can be a non-carbapenem treatment for canine infections of ESBL-producing *Escherichia coli*, *Klebsiella pneumoniae*, and *Proteus mirabilis*, but not for those of ESBL-producing *Enterobacter cloacae*.

## 1. Introduction

In recent years, infections caused by the multidrug-resistant extended-spectrum β-lactamase (ESBL)-producing Enterobacterales (ESBL-E) have become a serious issue in companion animals, as well as in humans [1,2]. In Japan, 15.5% and 34.0% of clinical isolates of *Escherichia coli* and *Klebsiella pneumoniae*, respectively, from companion animals were confirmed to be ESBL producers [3]. Carbapenems are frequently used for the treatment of ESBL-E infections in human medicine, but there is a risk of developing carbapenem-resistant Enterobacterales (CRE) [4,5]. Although the true prevalence of CRE in companion animals is unknown, there have been several reports on CRE isolation in dogs and cats worldwide [6,7]. Such prevalence of CRE in companion animals represents not only a serious concern in veterinary medicine but also a potential public health threat due to its transmission to surrounding people through close contact [8]. Therefore, the search for alternative drugs to treat ESBL-E infections is a high-priority issue in veterinary medicine.

Flomoxef (FMX) is an oxacephem antibiotic that is resistant to degradation by ESBLs because of its characteristic structure, with a methoxy group at the 7S position [9]. In human medicine, flomoxef is an effective alternative to carbapenems for the treatment of ESBL-E infections [10,11,12]. Furthermore, our previous studies confirmed high FMX susceptibility in ESBL-E derived from companion animals [13,14]. These findings suggest that FMX may be a potential alternative to carbapenems in companion animal medicine. However, there are insufficient reports on the pharmacokinetics (PK) of FMX in dogs, and a regimen of FMX for ESBL-E infections in dogs has yet to be established.

In recent years, PK–pharmacodynamics (PD) analysis with Monte Carlo Simulation (MCS) has been applied to explore available dosing regimens of antimicrobial drugs [15,16]. MCS can build large virtual populations by randomizing indices of PK and PD and thereby predict the likelihood of attaining a PK–PD target (probability of target attainment (PTA)) by dosage regimen [15,16]. The nonclinical PK–PD cutoff value is based on MCS analysis [17,18], which is a mathematical method that randomizes PK and PD indices by repeated random sampling. This allows for the estimation of the PTA of antimicrobial efficacy and the evaluation of the antimicrobial efficacy of the dosing regimen [17]. In this study, we first determined the PK parameters of FMX by administration experiments in healthy dogs. Next, based on the PK–PD relationship analyzed by MCS, we determined the nonclinical PK–PD cutoff values for dogs and proposed dosing regimens of FMX which can effectively treat canine ESBL-E infections.

## 2. Results

### 2.1. Linearity of Calibration Curves, Limit of Detection, and Limit of Quantification of High-Performance Liquid Chromatograph-Tandem Mass Spectrometry (LC–MS/MS)

In the LC-MS/MS method established in this study, high coefficients of determination (r^2^) were confirmed by the calibration curves of each dog (0.9986–0.9999). The limit of detection (LOD) and limit of quantification (LOQ) of FMX were determined as 0.005 and 0.01 µg/mL, respectively. The representative chromatogram of FMX (1 μg/mL) and an internal standard (IS) are shown in Appendix A.

### 2.2. PK Parameters of FMX in Dogs

The blood concentration–time curve and PK parameters of FMX when bolusly intravenous administrated at 40 mg/kg body weight are shown in Figure 1 and Table 1, respectively. The maximum blood concentration of FMX was 111.82 ± 19.60 μg/mL at 5 min after administration, which decreased exponentially.

### 2.3. Nonclinical PK-PD Cutoff Value of FMX in Dogs

The PTA values of FMX at the respective MIC when administered at 40 mg/kg body weight are shown in Figure 2. All regimens accomplished a PTA of more than 90% at an MIC of ≤0.5 μg/mL but not at an MIC of ≥16 µg/mL. Based on the obtained PTA values, the nonclinical PK–PD cutoff values of 40 mg/kg at every 12 h (q12h), 8 h (q8h), and 6 h (q6h) were set as ≤0.5, ≤2, and ≤8 µg/mL, respectively. Likewise, these values of 30 and 50 mg/kg were estimated as ≤0.25, ≤2, and ≤4 µg/mL and ≤0.5, ≤2, and ≤8 µg/mL, respectively. The PTA values of all doses are summarized in Appendix A.

### 2.4. CFR of FMX for ESBL-E Infections in Dogs

Table 2 summarizes the CFR values of a 40 mg/kg dose, which was obtained by estimating based on the MIC distribution of wild-type isolates of ESBL-E. Considering the estimated CFR, the regimens of 40 mg/kg q8h and q6h were optimal, and that of 40 mg/kg q12h was moderately successful for dogs infected with ESBL-*Escherichia coli*, -*K. pneumoniae*, and -*P. mirabilis*. In contrast, none of regimens achieved a CFR <80% for ESBL-*Enterobacter cloacae*-infected dogs.

## 3. Discussion

While various alternatives to carbapenems have been investigated in humans, few have been studied in companion animals such as canines. This study is the first to report the usefulness of FMX, an oxacephem antibiotic used in humans, against ESBL-E infections in dogs, based on a PK–PD approach. 

Antimicrobial susceptibility breakpoints are essential indicators for appropriate antimicrobial therapy. Hamada et al. [19] previously proposed nonclinical PK/PD breakpoints of FMX for humans, but those for dogs have not yet been established. In this study, we attempted to establish canine-specific nonclinical PK–PD cutoff values by using MCS analysis. Our data showed that nonclinical PK–PD cutoff values in dogs increase with shorter dosing intervals, irrespective of dose, as previously reported in humans [20]. In addition, the nonclinical PK–PD cutoff values of 40 mg/kg are higher than the MIC90 of ESBL-*K. pneumoniae* and -*P. mirabilis* (1 µg/mL each) and that of ESBL-*Escherichia coli* (4 µg/mL) [13,14] when administered at q8h and q6h, respectively. These data were supported by our other findings that the mean blood concentration of FMX at 40 mg/kg exceeded MIC90 of these bacterial species for 150–180 min. In addition, the CFR simulated in this study suggests that the q6h and q8h regimens of FMX are appropriate for the treatment of ESBL-*Escherichia coli*, -*K. pneumoniae,* and -*P. mirabilis* infections in dogs. The similar dosing intervals were proposed to achieve bactericidal concentrations against ESBL-E infections as found in human patients based on PK–PD simulations [20]. These findings in our study indicate that FMX administration at shorter dose intervals can be an alternative treatment for ESBL-*Escherichia coli*, -*K. pneumoniae*, and -*P. mirabilis* infections in dogs.

In contrast, all of the nonclinical PK–PD cutoff values calculated in this study were lower than the FMX MIC90 for ESBL-*Enterobacter cloacae* (>256 µg/mL) [14], possibly because the mean blood concentrations even at 5 min did not exceed the MIC90. This finding supports the finding that the CFR for ESBL-*Enterobacter cloacae* was not even moderately successful. It is known that *Enterobacter cloacae* has an inducible chromosomal AmpC β-lactamase, which can be induced by cephamycins, including oxacephems [21,22]. Therefore, FMX is unlikely to be a candidate drug for ESBL-*Enterobacter cloacae* infections in dogs, although infection with ESBL-*Enterobacter cloacae* is less prevalent in companion animals [23].

The optimum dose of FMX in dogs has not yet been established. In this study, we adopted 40 mg/kg per dose, according to the human dosage (i.e., a maximum of 37.5 mg/kg body weight four times a day), and investigated the blood PK of FMX in dogs when bolusly administered at this dose. In addition, our simulation implied that nonclinical PK–PD cutoff values of 40 mg/kg were higher than those of 30 mg/kg and were equivalent to those of 50 mg/kg. Mitsuzono et al. [24] estimated that the no-observed-effect level of FMX in dogs is 200 mg/kg/day based on a 6-month intravenous toxicity study. Therefore, we believe that the dosing regimens in this study (40 mg/kg q12h, q8h, and q6h) are the most reasonable from the viewpoint of efficacy and safety.

The results revealed similar values for elimination half-life and clearance per body weight compared with those in healthy human subjects, 44.2–46.2 min and 15.14 L/h, respectively [19,25]. This implies that the elimination rate of FMX in dogs is comparable to that in humans, although the protein binding rate of FMX in dogs is much lower (8%) [26], compared with humans (36.2%) [27]. In addition to such animal species differences, interindividual variability is the main factor confounding PK parameter [28]. After entering systemic circulation, the distribution of drugs can be affected by patient-specific pathophysiological characteristics including the rate of blood flow to the tissue [28]. In the elimination process, Kimura et al. [26] reported that almost all (97.7%) of FMX can be renally excreted in dogs, implying that the elimination of the drug greatly depends on renal function. In this study, PK parameters were calculated using healthy beagle dogs and may be different from those in patients with renal dysfunction, as confirmed in a human study [29]. Hamada et al. [26] reported that the PK/PD cutoff values of FMX can be inversely correlated with creatinine clearance in human patients. These findings would emphasize the need to adjust the dose of FMX according to pathophysiological characteristics and renal function in dogs.

There are several limitations in this study. First, only five dogs were used to estimate the PK parameters. Tam et al. [30] previously reported that a sample population size of ≥10 and ≥50 was found to be necessary to estimate the area under the concentration–time curve (AUC) and the standard deviation of the AUC, respectively, by using MCS. We could not include too many dogs because our university’s Animal Use Committee placed importance on reducing experimental animals. Second, the PK parameters of FMX are also different among body tissues [20], underlining the need to investigate the parameters in sites other than blood in dogs. Third, the correlation between drug concentration and antibacterial effect was not assessed because only the MIC was used as the PD parameter in this study. Further PD parameters (e.g., maximum antibacterial effect (E_max_) and drug concentrations at 50% E_max_) [31] should be considered to clarify the more accurate efficacy of FMX in dogs.

## 4. Materials and Methods

### 4.1. Animals

The animal experiments in this study were conducted under an ethics-committee-approved protocol in accordance with the Tottori University Animal Use Committee (Approval No. 19-T-17). Five beagle dogs were used in this study (four males and one female, aged 6.2 ± 1.8 years and weighing 13.6 ± 1.7 kg, SHIMIZU Laboratory Supplies Co., Ltd., Kyoto, Japan). The dogs were individually housed in each cage and confirmed to be clinically healthy based on physical tests, blood tests, and image examination prior to the study. They did not receive any medications in the 6 months prior to the examination. They were fed the same commercial food (Aiken Genki, Unicharm Corporation, Tokyo, Japan) and were individually housed in separate cages in the same room at the experiment animal facility.

### 4.2. Drug Administration and Serum Sampling

A central venous catheter (Covidien Japan, Inc., Tokyo, Japan) was placed in the jugular vein under general anesthesia on the day before the drug administration. Anesthesia was induced by intravenously administering propofol (4 mg/kg body weight, Propoflo, DS Pharma Animal Health Co., Ltd., Osaka, Japan) and subsequently intubated with a cuffed endotracheal tube. The vaporizer was adjusted to deliver 2% isoflurane (ISOFLURANE Inhalation Solution, Mylan EPD G.K., Tokyo, Japan) at an oxygen flow rate of 2 L/min. The FMX formulation (Shionogi Co., Ltd., Osaka, Japan) was dissolved in water for injection (Nissin Pharmaceutical Co., Ltd., Yamagata, Japan) and was bolusly administered at 40 mg/kg through the radial skin vein. Three-milliliter blood samples were collected from a central venous catheter before administration and 2 mL at 5, 10, 30, 45, 60, 90, 120, 150, 180, 240, 300, and 360 min after administration. Serum samples were obtained after centrifugation at 1300× *g* for 10 min after coagulation and stored at −80 °C until analysis.

### 4.3. Determination of Serum Concentrations of FMX in Dogs

The calculation of FMX concentration in serum samples was outsourced to NDTS, Inc. (Hokkaido, Japan). Briefly, as an IS, 200 µL of latamoxef (LMX) sodium (Shionogi, Osaka, Japan) solution (1 µg/mL) was added to the same volume of serum. After 100 μL of 20% sulfosalicylic acid was added, the IS was mixed vigorously for 30 s and centrifuged at 12,000× *g* for 5 min. Then, 250 μL of supernatant was collected and mixed with 250 μL of 100 mM acetic acid solution. The mixture was subjected to solid-phase extraction using Oasis HLB (1 cc, 30 mg; Waters, Milford, MA, USA). After loading, each sample was washed with 1 mL of 20 mM aqueous acetic acid solution, followed by elution with 1 mL of methanol. The eluted solution was dried at 35 °C under a stream of nitrogen and then dissolved into 100 μL of methanol. LC–MS/MS was performed on a high-performance liquid chromatography-mass spectrometer (Prominence and LCMS-8045 tandem mass spectrometer, Shimadzu Corporation, Kyoto, Japan). Separation by high-performance liquid chromatography was performed using two solutions, mobile phase A, 10 mM ammonium formate solution, and mobile phase B, 10 mM ammonium formate plus methanol, with the following gradient conditions: 5% (0 min); 40% (6 min); 100% (8 min); 100% (10 min); then 5% (10.5 min). After 5 µL of the sample was injected, target molecules were separated on a C18 reversed-phase column (Cadenza CD-C18, 3.0 mm i.d. × 150 mm, intact, Kyoto, Japan), which was controlled at a temperature under 40 °C. Mass spectrometry was performed in electrospray ionization (positive) and multiple monitoring reaction mode at a capillary voltage of 4.5 Kv, source (DL) temperature of 250 °C, nebulization gas of 180 L/hr, and drench gas of 10 L/min. LMX was detected at a monitor ion of m/z = 521 > 137 and collision energy of 27 V, and FMX was detected at an m/z = 497 > 137 and collision energy of 26 V. The area under the peak was determined by the analytical software LCMS solution ver. 3 (Shimadzu Corporation, Kyoto, Japan). The FMX concentration in each sample was calculated using a calibration curve with the serum obtained before drug administration, to which a known concentration of FMX sodium (Shionogi Co., Ltd., Osaka, Japan) had been added. 

### 4.4. Validation of the LC-MS/MS Method

The linearity of calibration curves was assessed by calculating r^2^ values in each dog. The LOD and LOQ were determined by blank determination [32]. Briefly, the mean concentration and the standard deviation of the blank are calculated by measuring replicates of the blank sample. The LOD was expressed as the analyte concentration corresponding to the sample blank value plus three standard deviations, and the LOQ was the analyte concentration corresponding to the sample blank value plus ten standard deviations [32]. 

### 4.5. Calculation of PK Parameters

The PK parameters from the non-compartment model were calculated based on the total blood concentrations of FMX measured in the five dogs using the PK package (ver. 4.0.3) of R software ver. 4.0.3 [33].

### 4.6. Monte Carlo Simulation

MCS was performed using commercial software (Oracle Crystal Ball version 11.1.2.4.850, Kozo Keikaku Engineering Inc., Tokyo, Japan) to calculate PTA based on the PK and PD parameter of FMX at a 40 mg/kg bolus dose at q12h, q8h, and q6h. Based on log-normally distributed PK parameters, 10,000 virtual patients were generated for each dosing regimen to construct drug serum concentration–time profiles. The percentage of time when the unbound drug concentration was above the MIC (fTAM), based on the serum protein binding rate of 8% [26], was employed as the PK–PD index to determine the optimal dosing regimen. The PK–PD target value was set as ≥40% according to a previous study [34]. 

The nonclinical PK–PD cutoff values of FMX were calculated as the highest MIC that achieved a PTA of ≥90% [18,35] when administered at 40 mg/kg. In addition, these values at 30 and 50 mg/kg q12h, q8h, and q6h were simulated by extrapolating the PK parameters of a 40 mg/kg bolus dose, except for the AUC, which was derived by dividing each total dose by the clearance. 

The cumulative fraction of response (CFR) of 40 mg/kg was calculated based on the wild-type MIC distribution, of which FMX in ESBL-E (*Escherichia coli*, *K. pneumoniae*, *P. mirabilis*, and *Enterobacter cloacae*) isolates from companion animals were determined in the previous studies [13,14]. A regimen with a CFR of ≥90% is defined as optimal, and a regimen with a CFR of 80–90% is defined as moderately successful [36].

## 5. Conclusions

In conclusion, we calculated the nonclinical PK–PD cutoff values of FMX at 40 mg/kg body weight q12h, q8h, and q6h by using MCS and estimated the CFR by considering the MIC distribution in wild-type isolates of ESBL-E. Our findings suggested that q8h and q6h dosage regimens of 40 mg/kg FMX are effective non-carbapenem treatment options for infections with ESBL-producing *Escherichia coli*, -*P. mirabilis*, and -*K. pneumoniae*. However, ESBL-producing *Enterobacter cloacae* infection in dogs cannot be treated with FMX. However, these findings have not yet been validated in dogs infected with ESBL-E, and therefore, further clinical studies are required to confirm the true efficacy of FMX.

## Figures and Tables

**Figure 1 ijms-25-01105-f001:**
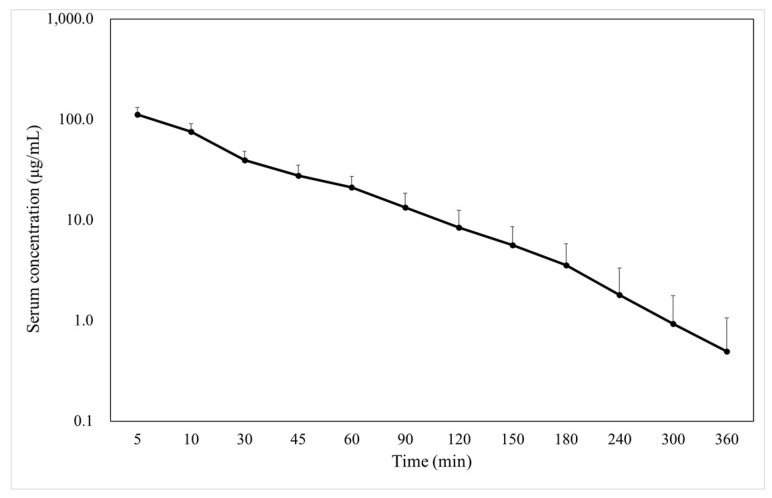
Semilogarithmic graph of total blood concentration of FMX in dogs administered at 40 mg/kg body weight (mean ± standard deviation, *n* = 5).

**Figure 2 ijms-25-01105-f002:**
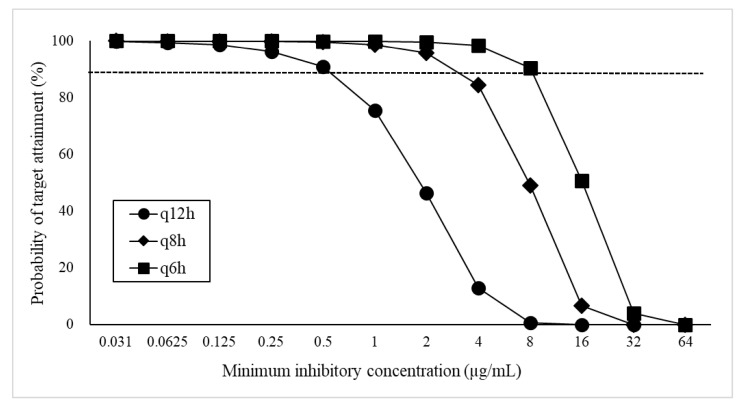
Probability of target attainment (PTA, %) at each minimum inhibitory concentration following intravenous administration of 40 mg/kg FMX. A broken line means a PTA of 90%.

**Table 1 ijms-25-01105-t001:** Pharmacokinetic parameter estimates after single-dose intravenous administration of FMX (40 mg/kg body weight) to dogs (*n* = 5).

Parameters (Unit) ^1^	Mean Values (Standard Deviation)
AUC (mg·h/L)	134.61 (11.3) ^2^
MRT (h)	1.10 (0.20)
T1/2 (h)	0.76 (0.14)
CL (L/h)	2.97 (0.33)
Vd (L)	3.27 (0.61)

^1^ AUC, area under the concentration–time curve; MRT, mean residence time; T1/2, elimination half-life; CL, total body clearance; Vd, volume of distribution. ^2^ In AUC, the coefficient of variation (%) was described in the parentheses.

**Table 2 ijms-25-01105-t002:** CFR of FMX when administered at 40 mg/kg body weight in dogs against ESBL-producing Enterobacterales.

Regimens	CFR (%)			
	*Escherichia coli*	*Klebsiella pneumoniae*	*Proteus mirabilis*	*Enterobacter cloacae*
q12h	83.63	87.12	87.85	23.12
q8h	91.32	92.40	96.22	50.17
q6h	93.26	94.13	98.57	65.28

## Data Availability

The data presented in this study are available in the article.

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
