# Peer review of "Pharmacokinetic and Pharmacodynamic Analysis of the Oxacephem Antibiotic Flomoxef against Extended-Spectrum β-Lactamase-Producing Enterobacterales from Dogs"

_ijms, 2024, doi:10.3390/ijms25021105_

Round 1

Reviewer 1 Report

Comments and Suggestions for Authors

In the manuscript, the authors conducted experimental and simulational study to determine the appropriate dosage of FMX in dogs with ESBL-E infections. Five dogs were treated with FMX in a dose of 40 mg/kg, and the serum FMX concentrations were measured to give its pharmacokinetics (Fig. 1 and Table 1) which is the most valuable result of this work. Monte Carlo Simulation were further employed using the parameters from current and previous experiments to estimate the appropriate dosing regimens of FMX. Overall, the procedure and results were sound, and the obtained insights are helpful for the clinic application of FMX. The manuscript can be accepted for publication after revision:

Major:

(1) It would be valuable to directly measure the ESBL in dogs.

(2) It would be better to delete "using Monte Carlo simulation" from the title.
Otherwise, some audiences may misunderstand the manuscript to be a simulation work, and thus reduce the impact of the article.
Actually, the experimental results on dogs (Fig. 1 and Table 1) is the most valuable ones,
while the simulation results (Fig. 2) are just data-processing issues.

Minor:

(3) L69: "which decreased gradually" is more exactly "which decreased exponentially".

Comments on the Quality of English Language

Minor revision may be made to improve the manuscript.

Author Response

Dear Reviewer 1:

We are grateful to Reviewer 1 for the critical comments and useful suggestions that have helped us to improve our paper. As indicated in the responses that follow, we have taken all these comments and suggestions into account in the revised version of our paper as much as possible.

Major:
Comment #1:

It would be valuable to directly measure the ESBL in dogs.
Response #1:

Thank you very much for your comment. The other investigators previously investigated the prevalence of ESBLs in Escherichia coli and Klebsiella pneumoniae isolates from companion animals in Japan. The finding and reference were described as follows:

(Line 36-38)

In Japan, 15.5% and 34.0% of clinical isolates of Escherichia coli and Klebsiella pneumoniae, respectively, from companion animals were confirmed to be ESBL producers ().

(Line 294-296)

Maeyama, Y.; Taniguchi, Y.; Hayashi, W.; Ohsaki, Y.; Osaka, S.; Koide, S.; Tamai, K.; Nagano, Y.; Arakawa, Y.; Nagano, N. Prevalence of ESBL/AmpC genes and specific clones among the third-generation cephalosporin-resistant Enterobacteriaceae from canine and feline clinical specimens in Japan. Vet. Microbiol. 2018, 216, 183-189.

Comment #2:

It would be better to delete "using Monte Carlo simulation" from the title. Otherwise, some audiences may misunderstand the manuscript to be a simulation work, and thus reduce the impact of the article.
Actually, the experimental results on dogs (Fig. 1 and Table 1) is the most valuable ones, while the simulation results (Fig. 2) are just data-processing issues.

Response #2:

I am very grateful to you for providing the valuable comment. According to your comment, the title was revised as follows.

Pharmacokinetic and pharmacodynamic analysis of the oxacephem antibiotic flomoxef against extended-spectrum β-lactamase-producing Enterobacterales from dogs

Minor:
Comment #3:

L69: "which decreased gradually" is more exactly "which decreased exponentially".

Response #3:

According to your comment, the corresponding sentence was revised as follows.

(Line 79-80)

The maximum blood concentration of FMX was 111.82 ± 19.60 μg/mL at 5 min after administration, which decreased exponentially.

We also revised our manuscript according to Reviewer 2’s comments and carried out necessary revisions to improve the quality of the manuscript. We greatly appreciate your help concerning the improvement of this paper. We hope that the revised manuscript is now acceptable for publication.

Yours very sincerely,

Kazuki Harada, DVM, Ph. D.

Reviewer 2 Report

Comments and Suggestions for Authors

The authors conducted a PK study of flomoxef (FMX) in healthy dogs to establish an appropriate treatment regimen. In the study, 5 dogs received a single dose of 40 mg/kg, and the dosage regimen was estimated using commercial software.

However, several concerns have been identified:

1-Please, clarify how the sample size (n) was calculated. Provide details on the statistical considerations and rationale for selecting only 5 dogs.

2-The authors selected a dose of 40 mg/kg, but it is unclear why this specific dose was chosen. Given that the optimal dose of FMX for dogs has not been established, it is recommended to consider multiple doses. Additionally, the authors claimed a lower protein binding site in dogs (8%) than humans (36.2%). Still, it was not confirmed as the total concentration, rather than the actual free concentration, was measured. It is recommended to investigate the impact of protein binding on free drug levels, particularly to Time above minimal inhibitory concentration (TAM).

3-The authors included some previously published data (in vitro determination of MIC of FMX) as current results (line 78). Data previously published cannot be included in the results section.

4- Matsumura et al (Antimicrob Agent Chemother 2015) reported that the MIC90 in patients was <1 mg/L, suggesting a MIC is lower than the point suggested in this study. In addition, according to Hamada Y (antibiotics 2022), the optimal dose varies according to renal function. Provide information on the renal clearance of the dogs, given the variability in the optimal dose based on renal function.

5-Include a graphical representation of concentration-time profiles. Express AUC data as mean and %CV, not with SD. Incorporate other essential PK parameters such as Cmax and Tmax.

6- The pharmacodynamic study was not included because there was not studied the correlation between drug concentration and the antibacterial effect in dogs. Additionally, the estimation of EC50 (concentration producing 50% of the maximum effect) or Emax (maximum effect) was not included. Please clarify or correct this omission.

7-The PK model has not been validated in sick dogs. It is supposed that the interest in this topic stems from the authors’ previous experience with infected dogs. Did the authors treat infected dogs with this antibiotic regimen?

8-The discussion claimed that the breakpoint for FMX has not been established, but a reference (Hamada Y et al. antibiotics 2022) suggests otherwise. Please clarify.

9-The study did not evaluate renal clearance or toxicity. Given the variability in optimal doses on renal function, it is recommended to include an assessment of renal clearance in the study.

10-The validation of the LC-MS/MS should be included in the methodology, covering aspects of sensitivity and specificity. Provide details on the validation parameters to ensure the reliability of the analytical method.

11- The study did not explore inter-individual variability. Consider including an analysis of factors contributing to individual variation in drug response.

Author Response

Dear Reviewer 2:

We are grateful to Reviewer 2 for the critical comments and useful suggestions that have helped us to improve our paper. As indicated in the responses that follow, we have taken all these comments and suggestions into account in the revised version of our paper as much as possible.

Comment #1:

Please, clarify how the sample size (n) was calculated. Provide details on the statistical considerations and rationale for selecting only 5 dogs.

Response #1:

We sincerely receive your comment. Tam et al. previously reported that a sample population size of ≥10 and ≥ 50 was found to be necessary to estimate AUC and standard deviation of AUC, respectively, by using MCS. In this study, we could not satisfy such a sample size because the Animal Use Committee of our university strongly requested the reduction of the sample size as much as possible. We would appreciate it if you could accept these circumstances. To explain this background, the following sentences were added.

(Line 169-173)

Tam et al. [30] previously reported that a sample population size of ≥10 and ≥ 50 was found to be necessary to estimate the area under the concentration-time curve (AUC) and standard deviation of AUC, respectively, by using MCS. We could not satisfy such many dogs because our university's Animal Use Committee placed importance on reducing experimental animals.

In addition, the following reference was added.

  1. Tam, V.H.; Kabbara, S.; Yeh, R.F.; Leary, R.H. Impact of sample size on the performance of multiple-model pharmaco-kinetic simulations. Antimicrob. Agents Chemother. 2006, 50, 3950-3952.

Comment #2:

The authors selected a dose of 40 mg/kg, but it is unclear why this specific dose was chosen. Given that the optimal dose of FMX for dogs has not been established, it is recommended to consider multiple doses. Additionally, the authors claimed a lower protein binding site in dogs (8%) than humans (36.2%). Still, it was not confirmed as the total concentration, rather than the actual free concentration, was measured. It is recommended to investigate the impact of protein binding on free drug levels, particularly to Time above minimal inhibitory concentration (TAM).

Response #2:

Thank you very much for your comment. We calculated PK/PD cutoff values doses of 30 and 50 mg/kg, in addition to 40 mg/kg dose. As the results, PK/PD cutoff values of 40 mg/kg were higher than those of 30 mg/kg and were equivalent to those of 50 mg/kg. Thus, we believe that the dosing regimens in this study (40 mg/kg q12h, q8h, and q6h) are the most reasonable from the viewpoint of efficacy and safety. To clarify this point, the following sentences were added.

(Line 94-97)

Likewise, these values of 30 and 50 mg/kg were estimated as ≤ 0.25, ≤ 2, and ≤ 4 µg /mL and ≤ 0.5, ≤ 2, and ≤ 8 µg /mL, respectively. The PTA values of all doses were summarized in Supplementary Table S1.

(Line 146-148)

In addition, our simulation implied that nonclinical PK–PD cutoff values of 40 mg/kg were higher than those of 30 mg/kg and were equivalent to those of 50 mg/kg.

(Line 254-258)

The nonclinical PK–PD cutoff values of FMX were calculated as the highest MIC that achieved a PTA of ≥90% [18,36] when administered at 40 mg/kg. In addition, these values at 30 and 50 mg/kg q12h, q8h, and q6h were simulated by extrapolating PK parameters at a 40 mg /kg bolus dose, except for the AUC, which was derived by dividing each total dose by the clearance.

In addition, all of the PTA and PK/PD cutoff values were summarized in Supplementary Table S1.

In this study, total drug concentrations were taken into account when calculating PK parameters, whereas free drug concentrations when calculating PTA values. To clarify this point, the following revisions were made.

(Line 81-82)

Figure 1. Semilogarithmic graph of total blood concentration of FMX in dogs administered at 40 mg/kg body weight (mean ± standard deviation, n = 5).

(Line 242-244)

The PK parameters from the non-compartment model were calculated based on the total blood concentrations of FMX measured in five dogs using the package PK (ver. 4.0.3) of R software [34].

Comment #3:

The authors included some previously published data (in vitro determination of MIC of FMX) as current results (line 78). Data previously published cannot be included in the results section.

Response #3:

We strongly agree with your comment. To respond to your comment, the corresponding data was moved to the discussion section after condensing, as follows.

(Line 125-127)

These data were supported by our other findings that the mean blood concentration of FMX at 40 mg/kg exceeded MIC90 of these bacterial species for 150–180 min.

(Line 135-137)

In contrast, all of the nonclinical PK–PD cutoff values calculated in this study were lower than FMX MIC90 for ESBL-Enterobacter cloacae (>256 µg/mL) [14], possibly because the mean blood concentrations at even 5 min did not exceed the MIC90.

Comment #4:

Matsumura et al (Antimicrob Agent Chemother 2015) reported that the MIC90 in patients was <1 mg/L, suggesting a MIC is lower than the point suggested in this study. In addition, according to Hamada Y (antibiotics 2022), the optimal dose varies according to renal function. Provide information on the renal clearance of the dogs, given the variability in the optimal dose based on renal function.

Response #4:

Thank you very much for the valuable comment. Kimura et al. (1987) previously reported that 97.7% of flomoxef was renally excreted in dogs. Thus, we believe that the excretion of flomoxef can be greatly affected by renal function. Based on this point and Comments #9 and 11, we added the following sentences.

(Line 157-168)

In addition to such animal species differences, interindividual variability is a main factor confounding PK parameters [28]. After entering systemic circulation, the distribution of drugs can be affected by patient-specific pathophysiological characteristics including the rate of blood flow to the tissue [28]. In the elimination process, Kimura et al. [26] reported that almost all (97.7%) of FMX can be renally excreted in dogs, implying that the elimination of the drug greatly depends on renal function. In this study, PK parameters were calculated using healthy beagle dogs and may be different from those in patients with renal dysfunction, as confirmed in a human study [29]. Hamada et al. [26] reported that the PK/PD cutoff values of FMX can be inversely correlated with creatinine clearance in human patients. These findings would emphasize the need to adjust the dose of FMX according to pathophysiological characteristics and renal functions in dogs.

Comment #5:

Include a graphical representation of concentration-time profiles. Express AUC data as mean and %CV, not with SD. Incorporate other essential PK parameters such as Cmax and Tmax.

Response #5:

Thank you very much for your comment. We have already added a graphical representation of blood drug concentration-time in Table 1. If you indicate the other form of representation, we would be delighted to accept your indication. In addition, AUC data was described as mean and %CV in Table 1. The following footnote was added to Table 1.

(Line 88)

2 In AUC, the coefficient of variation (%) was described in the parenthesis.

We would like to deeply apologize for the mistakes that standard errors but not standard deviations were described in the previous version of Table 1. Thus, the data in the parenthesis of the parameters other than AUC were replaced with standard deviations.

In addition, FMX was bolus intravenously administrated in this study, and then Cmax and Tmax could not be calculated. We would appreciate it if you could consider this circumstance.

Comment #6:

The pharmacodynamic study was not included because there was not studied the correlation between drug concentration and the antibacterial effect in dogs. Additionally, the estimation of EC50 (concentration producing 50% of the maximum effect) or Emax (maximum effect) was not included. Please clarify or correct this omission.

Response #6:

We sincerely receive your comment. As you indicated, we used only MIC as the PD parameter. Thus, your indication is considered to be one of the limitations of this study. To clarify this point, the following sentences were added.

(Line 176-180)

Third, the correlation between drug concentration and the antibacterial effect was not assessed because only MIC was used as the PD parameter in this study. Further PD parameters [e.g., maximum antibacterial effect (Emax) and drug concentrations at 50% Emax] [32] should be considered to clarify the more accurate efficacy of FMX in dogs.

Comment #7:

The PK model has not been validated in sick dogs. It is supposed that the interest in this topic stems from the authors’ previous experience with infected dogs. Did the authors treat infected dogs with this antibiotic regimen?

Response #7:

We sincerely receive your comment. As we described in the Introduction part, this study stemmed from the fact that FMX has high in vitro efficacy against ESBL-E and is highly safe for dogs, but not from our experiences. Thus, as you indicated, the present findings have to be further validated in clinical studies. To clarify this point, the following sentence was added.

(Line 272-273)

However, these findings have not yet been validated in dogs infected with ESBL-E, and thereby further clinical studies are required to confirm true efficacy of FMX.

Comment #8:

The discussion claimed that the breakpoint for FMX has not been established, but a reference (Hamada Y et al. antibiotics 2022) suggests otherwise. Please clarify.

Response #8:

Thank you very much for your valuable comment. In response to your comment, the following sentences were revised and added.

(Line 117-1119)

Antimicrobial susceptibility breakpoints are essential indicators for appropriate antimicrobial therapy. Hamada et al. [19] previously proposed nonclinical PK/PD breakpoints of FMX for humans, but those for dogs have not yet been established.

The further corresponding sentence was separately added. Please see Response #4.

Comment #9:

The study did not evaluate renal clearance or toxicity. Given the variability in optimal doses on renal function, it is recommended to include an assessment of renal clearance in the study.

Response #9:

We strongly agree with your comment. In response to your comment, the sentences to elucidate renal clearance were added. Please see Response #4.

Comment #10:

The validation of the LC-MS/MS should be included in the methodology, covering aspects of sensitivity and specificity. Provide details on the validation parameters to ensure the reliability of the analytical method.

Response #10:

We sincerely receive your comment. In response to your comment, the following sentences were added.

(Line 68-74)

2.1. Linearity of calibration curves, limit of detection, and limit of quantification of high-performance liquid chromatograph-tandem mass spectrometry (LC-MS/MS)

    High coefficients of determination (r2) were confirmed in the calibration curves of each dog (0.9986­–0.9999). The limit of detection (LOD) and limit of quantification (LOQ) of FMX were determined as 0.005 and 0.01 µg /mL, respectively. The representative chromatogram of FMX (1 μg/mL) and an internal standard (IS) is shown in the supplementary Figure S1.

(Line 233-239)

4.4 Validation of the LC-MS/MS method

The linearity of calibration curves was assessed by calculating r2 values in each dog. The LOD and LOQ were determined by blank determination [33]. Briefly, the mean concentration and the standard deviation of the blank are calculated by measuring replicates of the blank sample. LOD was expressed as the analyte concentration corresponding to the sample blank value plus three standard deviations and LOQ was the analyte concentration corresponding to the sample blank value plus ten standard deviations [33].

In addition, the representative chromatogram was added as the supplementary Figure S1.

Comment 11:

The study did not explore inter-individual variability. Consider including an analysis of factors contributing to individual variation in drug response.

Response 11:

We strongly agree with your comment. In response to your comment, the sentences to elucidate inter-individual variability were added. Please see Response #4.

We also revised our manuscript according to Reviewer 1’s comments and carried out necessary revisions to improve the quality of the manuscript. We greatly appreciate your help concerning the improvement of this paper. We hope that the revised manuscript is now acceptable for publication.

Yours very sincerely,

Kazuki Harada, DVM, Ph. D.

Round 2

Reviewer 2 Report

Comments and Suggestions for Authors

I agree with the authors' response  and the paper has been improved. The paper is now ready for publication.